# Analysis of Psychosomatic Disorders According to Age and Sex in a Rural Area: A Population-Based Study

**DOI:** 10.3390/jpm12101730

**Published:** 2022-10-18

**Authors:** Elisabet Torrubia-Pérez, Silvia Reverté-Villarroya, José Fernández-Sáez, Maria-Antonia Martorell-Poveda

**Affiliations:** 1Nursing Department, Campus Terres de l’Ebre, Universitat Rovira i Virgili, 43500 Tortosa, Spain; 2Advanced Nursing Research Group, Universitat Rovira i Virgili, 43002 Tarragona, Spain; 3Pere Virgili Health Research Institute, Hospital de Tortosa Verge de la Cinta, Catalan Institute of Health, Carretera Esplanetes, 14, 43500 Tortosa, Spain; 4Unitat de Suport a la Recerca Terres de l’Ebre, Fundació Institut Universitari Per a la Recerca a l’Atenció Primària de Salut Jordi Gol i Gurina (IDIAPJGol), 43500 Tortosa, Spain; 5Nursing Department, Campus Catalunya, Universitat Rovira i Virgili, 43002 Tarragona, Spain

**Keywords:** psychosomatic disorders, sex, age, biopsychosocial, mental health, prevalence

## Abstract

Psychosomatic disorders can develop unevenly depending on certain health determinants; therefore, the aim of this study was to analyze the prevalence of psychosomatic disorders and the differences by age and sex in a rural area. We conducted an observational descriptive retrospective population study to determine the prevalence of 201 diagnoses of psychosomatic nature grouped into 25 diagnostic categories by sex and age groups. A total of 33,680 participants with a diagnosis of psychosomatic disorder were identified (64.6% women, 35.4% men). We found statistically significant differences based on sex in 13 of the 25 diagnostic categories previously defined. When we analyzed these categories by age, we found that women showed a higher probability, between 1.23 and 10.85 times, of suffering from most of these health issues. We also observed that the older the age group, the most often they had a diagnosis. Notably, more women seem to suffer from psychosomatic disorders when compared to men in the same situation. In most of these disorders, being of the female sex was a risk factor, and the older the participants, the greater the probability of developing a disorder.

## 1. Introduction

The psychosomatic model is based on the biopsychosocial model. Biopsychosocial model sustains that, regardless of the health problem, the biological, psychological, and social spheres of a person are affected [1,2]. Along these lines, psychosomatic disorders can result from the interaction of the individual with any of the three spheres mentioned above, including an emotional discomfort caused by situations in which individuals find themselves [3]. There are certain health determinants that can cause mental health issues [4] that are conditioned by the prevailing sociocultural inequalities in a given context [1,5].

The Western socio-cultural model is structurally patriarchal and inherently changes these determinants, participating in the possible development of certain health issues of marked psychosomatic nature. This is reflected in specific inequalities found in the incidence and prevalence of certain conditions according to sex, including mental health issues [5,6].

According to the 2017 Health Survey of Catalonia, despite the lower values obtained in mental well-being scales, 19.8% of women referred to suffering from anxiety or depression in Catalonia, compared to 10.3% of men [7,8]. Likewise, we observed a higher prevalence in women (42.3% versus 36.1%) of chronic illnesses, scoring higher in pain and discomfort: in 1 out of 5 men, and 1 out of 3 women. It is worth noting that this proportionality was higher in women in all age groups and the scores obtained in Terres de l’Ebre (a rural area in the south of Catalonia) were higher than the Catalan average [7,9]. Subsequent studies carried out in this health region showed higher rates of anxiety and depression when compared to other areas within the Catalonia region [10].

The latest report from the Health Survey of Catalonia (2020) highlights that nearly a quarter of the population of 15 years of age and older suffers from emotional distress, as reported by 17.3% of men compared to 32% of women, revealing a significant difference. As shown in these annual health surveys, the total cases of emotional distress are increasing, and reports confirm the trend that women experience it in a greater percentage [10].

Furthermore, studies carried out in Catalonia and the rest of Spain show that the parameters defined to study the perception of quality of life [11,12], positive self-perception of health status [10] and risk of suffering from a mental disorder (10% women vs. 5.3% men) affect women more negatively than men [12]. After confirming this vulnerability of women in different periods (2006, 2011 and 2017), a study analyzing mental health issues by autonomous region concluded the need to explore the individual and contextual causes that determine the significantly higher scores of psychiatric morbidity in women compared to men [8].

Given the exposed inequalities in terms of mental health determinants and issues, we must underline the importance of conducting research in this field from a gender perspective [13]. Studies related to mental health must contemplate this key variable in order to detect potentially modifiable determinants of these inequalities and contribute to their solution [14,15]. Analyzing psychosomatic issues with a gender perspective can provide unprecedented information on the morbidity of mental health disorders and their development [16,17].

Therefore, the aim of this study was to analyze the prevalence of psychosomatic disorders and their differences according to age and sex in a rural area.

## 2. Materials and Methods

### 2.1. Study Design

We conducted an observational descriptive retrospective population study, using clinical data extracted from the SIDIAP database (Information System for Research in Primary Care), a computerized clinical history management tool used by primary care centers of the Catalan Institute of Health (ICS, for its Catalan acronym). In March 2021, we extracted the information available for the last 10 years. The Strengthening the Reporting of Observational studies in Epidemiology (STROBE) guidelines were followed in this observational study.

### 2.2. Study Setting and Sampling

The study population consisted of men and women from the Terres de l’Ebre Health Region, a rural population found in the south of Catalonia (Spain). According to the 2020 demographic report, there are 179,574 inhabitants in this region. Terres de l’Ebre consists of four regions, with the following populations: Baix Ebre (78,011 inhabitants), Montsià (62,263), Ribera d’Ebre (21,870) and Terra Alta (11,430) [18].

The sample was selected through non-probability sampling, following the established inclusion and exclusion criteria. The inclusion criteria were: (1) age ≥ 18 years old, (2) active clinical history, (3) resident of the Terres de l’Ebre Health Region and (4) ≥1 diagnosis of a health issue among those selected for this study and coded as per the ICD-10-ES Diagnostic Coding Manual [19] in the clinical history of Primary Care (Annex 1). We excluded those living outside this region at the time of sample selection and those diagnosed before 2010.

After applying these criteria to the total population (N = 197,574), we created an initial database with 43,892 cases, which were subsequently reviewed and filtered with greater precision, resulting in a final sample of 33,680 cases (Figure 1).

### 2.3. Measures

We extracted sociodemographic variables from the SIDIAP database that helped characterize the profile of the participants: sex, date of birth, and place of residence (region).

The independent variables were sex and age. The latter was stratified by the following age groups: 18–24, 25–44, 45–64, and ≥65 years, according to the classification used by the National Institute of Statistics of Spain for the analysis of health determinants [20]. This age categorization allows an easier comparison between the results obtained in this study and the national level results.

Additionally, the variables of the study related to the diagnosis of health issues were defined based on the International Classification of Diseases ICD-10-ES [19]. Subsequently, we selected a total of 201 of the most representative diagnoses of the psychosomatic process that have a greater impact on the health determinants that derive from the sociocultural context of this study [4,14,15,17,21,22,23]. To effectively analyze how they are associated with sex, we grouped them into 25 diagnostic categories according to specific health issues (described in Appendix A).

Among these categories, we selected the following related to mental, behavioral and neurodevelopmental disorders: (1) depression, (2) persistent mood disorder, (3) anxiety, (4) stress, (5) conversion disorder, (6) somatoform disorder, (7) non-psychotic mental disorder, (8) eating disorder, and (9) sleep disorder.

Categories related to symptoms, signs and abnormal results of supplementary tests were (10) symptoms and signs affecting the emotional state, (11) symptoms and signs related to appearance and behavior, and (12) malaise and fatigue.

Categories related to health issues as a result of external factors: (13) abuse, neglect and other mistreatment, confirmed; (14) abuse, neglect and other mistreatment, suspected.

Additionally, lastly, categories related to factors affecting the health status and contact with healthcare services: (15) problems related to (PR) to employment and unemployment; (16) PR to housing and financial circumstances; (17) PR to social environment; (18) PR with parenting; (19) PR with primary support group, including family circumstances; (20) other psychosocial circumstances; (21) counseling related to attitude, behavior, and sexual orientation; (22) contact with health services for other medical concerns; (23) PR to lifestyle; (24) RP to difficulties in controlling their life; and (25) problems related to the care of a dependent person.

### 2.4. Data Analysis

We analyzed the data using statistical software IBM SPSS for Windows (Version 27.0). After recoding the variables to adapt them to our analysis, we used descriptive statistics to describe the sociodemographic information of the sample population and the characteristics of the most prevalent conditions found in this region. Sociodemographic variables were stratified according to age groups (18–24, 25–44, 45–64, ≥65) and were distributed by region.

We proceeded to verifying the normality of the distribution of the quantitative variables by parametric and categorical tests, and analyzed the dependence between variables by the Chi Square hypothesis contrast test to examine the bivariate associations between sex and diagnosis.

Significant associations were incorporated into the logistic regression, and we applied the odds ratio to determine the strength of the relationship between the diagnostic categories and sex. We adjusted the OR for age and the significance level top value < 0.05 for its analysis.

### 2.5. Ethical Considerations

This study was approved by the Research Ethics Committee of the Jordi Gol University Institute of Primary Care Research (Jordi Gol Institut Universitari d’Investigació en Atenció Primària—IDIAP Jordi Gol) with code 20/157-P in December 2020. We extracted completely anonymized data and used the information for the sole purpose of disclosing new scientific content.

## 3. Results

We analyzed a total of 33,680 cases that met the inclusion criteria of the database. Of these cases, 64.6% were women (n = 21,772) and the remaining 35.4% were men (n = 11,908).

The mean age of the participants included in the sample was 57.71 (standard deviation (SD) = 18.73), and it was slightly higher in women (58.71 ± 18.75 years) than in men (55.88 ± 18.57 years). Table 1 shows that, the older the participant, the higher is the prevalence of presenting psychosomatic disorders, with women predominating in all groups (*p* < 0.001).

Geographically, the distribution of cases by region was 45.8% in Baix Ebre, 36.7% in Montsià, 12.1% in Ribera d’Ebre, and 5.4% in Terra Alta, with a greater prevalence of women.

Table 2 shows that anxiety was the most prevalent diagnostic cluster (n = 18,311), followed by sleep disorders (n = 10,896) and depression (n = 8427). A high number of cases from the cluster stress (n = 1945), malaise and fatigue (n = 1848) and persistent mood disorders (n = 1213) were also recorded. Other categories, to a lesser extent, were present in a high volume of cases, although in some of these the representation is minimal, such as cases of suspected abuse, neglect and other mistreatment (n = 4) or problems related to parenting (n = 7).

After applying the Chi-Squared hypothesis contrast test to verify the association between sex and diagnosis, we identified statistically significant differences in 13 of the 25 established diagnostic groups (Table 2).

The diagnostic categories with a *p* value < 0.001 when comparing men and women were depression, persistent mood disorder, anxiety, eating disorder, sleep disorder, malaise and fatigue, confirmed abuse, neglect and other types of mistreatments, support issues with the primary support group and problems with dependency care. Other significant groups were non-psychotic mental disorder (*p* = 0.005), signs and symptoms affecting the emotional state (*p* = 0.002), problems related to housing and financial circumstances (*p* = 0.018), and problems related to lifestyle (*p* = 0.002). 

We performed a logistic regression using sex as the dependent variable, which showed a that the female sex is significantly associated with some of the health issues analyzed in this study (aOR = 1.28; 95% CI: 1.22–1.28; *p* < 0.001) (Table 3). Other statistically significant categories were confirmed abuse, neglect, and other mistreatment (aOR = 10.85; 95% CI: 2.61–45.14; *p* < 0.001) and eating disorder (aOR = 3.58; 95% CI: 2.47–5.20; *p* < 0.001). We also observed an inverse association between diagnosis and female sex in non-psychotic mental disorders (aOR = 0.22; 95% CI: 0.07–0.70; *p* < 0.001) and sleep disorders (aOR = 0.82; 95% CI: 0.78–0.86; *p* < 0.001), signs and symptoms affecting the emotional state (aOR = 0.79; 95% CI: 0.68–0.92; *p* = 0.002) and PR to lifestyle (Aor = 0.47; 95% CI: 0.28–0.79; *p* = 0.004).

Regarding the age groups analyzed by sex and diagnosis, we found diverse results. In the categories of persistent mood and anxiety disorders, we found that the risk of developing these health issues was also increased in women, with significant aOR in all age groups. We observed similar results in the categories of depression and malaise and fatigue, except for the first age group (18–24) in which we obtained a *p* value above 0.05.

The category with the highest aOR values is eating disorders in all age ranges—except for those over 65 years—in which *p* value is not significant. In the other age groups, significance was observed, and it is worth noting that the age group 18–24 years showed the highest correlation strength (aOR = 6.11; 95% CI: 2.83–13.18; *p* < 0.001).

Regarding the category abuse, neglect and other mistreatment, we found that there are no registered cases of people of 24 years of age or less, but it is between 25 and 44 years that being a woman greatly determines the probability of suffering from this problem (aOR = 14.82; 95% CI: 1.99–110.37; *p* < 0.009). We also noted a high statistical score in the age group 45–64 years (aOR = 5.54; 95% CI: 0.71–43.34; *p* = 0.103), although without any statistical significance. No cases were registered in the database of people over 65 years of age within this diagnostic category.

When analyzing the PR to the primary support group and the PR to the care of a dependent person, we observed a positive probability trend as age increases, but not in younger stages. 

Regarding the categories with inverse association with the female sex, we detected specific significance values (over 65 years of age with PR to lifestyle aOR = 0.26: 95% CI: 0.10–0.72; *p* = 0.010) and a strong correlation between male sex and the category signs and symptoms affecting the emotional state in early life stages (18–24 years: aOR = 0.34; 95% CI: 0.14–0.80; *p* = 0.013; 25–44 years old: aOR = 0.53; 95% CI: 0.36–0.81; *p* < 0.003). The analysis by age and sex in the category non-psychotic mental disorder did not show any significance at any age group. Although the analysis by sex suggests an association between sleep disorders and the male sex, when incorporating age as variable, we observed that the only significant age group is 45–64 years old (aOR = 1.10, 95% CI: 1.01–1.21; *p* = 0.038), associated with the female sex.

## 4. Discussion

Psychosomatic disorders often remain blurred in the nosology of mental health issues [4]. Therefore, we have selected those that reflect the biological, psychological and social interactions of the health process to a greater extent and within this scope. The aim of this study was to analyze the prevalence of psychosomatic disorders and their differences according to age and sex in a rural area. The case distribution after the analysis of the data suggests that there is substantial differential morbidity based on sex, and, additionally, a progressive increase in the number of cases with a diagnosis of a psychosomatic condition as the age of the participants increases.

The application of the sample selection criteria resulted in a total of 33,680 cases of men and women diagnosed with the selected conditions, with a prevalence of 18.75% in the region. It is considered that other conditions with a similar prevalence in Terres de l’Ebre, such as hypercholesterolemia (19.7%) or lower (chronic migraine, 13.5%; diabetes, 8.5%) [10], are frequent, suggesting that psychosomatic disorders are also prevalent in the region.

The prevalence value of 18.75% was extracted from the latest population census in the Terres de l’Ebre Health region [18]. However, if this percentage is broken down by sex, the prevalence of women resident in Terres de l’Ebre with these issues is notably higher (24.42%) than that of men who are also residents of the same region (13.1%). This distribution is consistent with the last National Health Survey conducted in 2017, where global psychic morbidity scored a total of 19.1%, of which 22.8% were women and 14.6% were men [8], following the pattern already observed. That same year, the prevalence of poor mental health (not necessarily diagnosed) was 23.4% in women and 15.6% in men. We noted that these scores are equally higher for women in all age groups [24]. 

In studies that analyze the prevalence of mental health issues (which include most of those selected for this study), similar prevalence rates are observed, and even slightly lower than those resulting from our research. In 2006, Catalonia recorded a morbidity rate related to mental health of 21.5% in women, and 14% in men; while in 2011, the values increased to 24.3% in women and 14.7% in men [8]. At country level, the figures for 2006 and 2011 are 26.7% women vs. 15.3% men, and 26.1% women vs. 17.5% men, respectively, which are slightly higher than the values for the autonomous region of Catalonia and those obtained in our study. It is worth noting that the parameters established in our study differ slightly, and that these studies cover a greater number of conditions.

Incorporating a psychosomatic perspective to these disorders typically categorized as mental health issues is extremely important (in this study, from C01F32.0 to C01F51.19) due to the theoretical basis of how life conditions and individual situations can impact this sphere [24,25]. These situations, in addition to health issues of, typically, a psychosocial nature (in the study, from code C01R45.0) should be analyzed intersectionally in order to understand how the different axes of inequality interact.

Due to this intersectionality, considering age as a variable is essential when studying health issues that have a clear psychosocial and cultural influence. In this study, the mean age of the participants in the sample (58.71 women vs. 55.88 men) shows a high standard deviation both in women (SD = 18.75) and in men (SD = 18.57), which suggests that a large percentage of the population has been diagnosed with psychosomatic disorders. However, it is evident that the number of cases is greater in the older age groups. Furthermore, this prevalence is gradually increasing in all regions, with the largest number of cases with psychosomatic diagnoses being the age group over 65 years. This is consistent with previous research reflecting the relationship between sex, age and mental health [26,27].

This work suggests that, out of the total number of cases, the prevalence increases from 59.06%, in women between 18 and 24 years old, to 68.13%, in women over 65 years of age, indicating that the number of women with psychosomatic diagnoses increases with age. These findings are consistent with other studies, which show that at older ages the prevalence of mental health problems tends to increase, but sex-related differences tend to be more pronounced at younger ages [27,28]. However, some studies obtained different results, which would indicate that the greater risk of developing mental health issues is during the earliest stage of adulthood [27].

It should be noted that, in this study, we categorized the diagnoses using the generic categories described in the ICD-10-ES [19], as it is a recognized international classification and describes a greater number of generic categories used in clinical practice, unlike other studies [10,23,29]. Within this categorization, when measuring associations, we observed different trends by age and sex in the various diagnoses included in this study. Thus, we see that certain categories, such as eating behavior disorders (EDs) and signs and symptoms affecting the emotional state, show a diagnostic trend in emerging ages. On the contrary, in the PR to primary support group, that is, the social environment, lifestyle, or depression, a greater disparity is observed between the cases according to sex at older ages.

Based on the latest Health Survey of Catalonia, 10.6% of the population over 15 years of age suffer from major depression or severe major depression (13.7% of women and 7.4% of men). This percentage increases with age, especially after 75 years, which corresponds to a global 18.8% [10]. In this study, we observed that the increasing trend is consistent and that, after comparing the cases by age groups—except in the group of 18–24 years—being a woman implies a greater probability of suffering from depression (*p* < 0.001). 

Depression is clearly influenced by psychosocial factors, given that differential morbidities are explained by the context and the living conditions that act as health determinants [30]. In this study, the female sex was a risk factor for anxiety in all age groups (aOR = 1.38–1.63; *p* < 0.001), which is consistent with the existing literature. Anxiety is also one of the most prevalent mental health issues and, like depression, considerably affects the health determinants within a specific context [31]. Depression and anxiety both have been associated with chronic illnesses, which also show differences by sex [32], which is in line with the results obtained in this study. 

In the literature, stress is often regarded as an important health issue in the general population, which is often combined with anxiety. Although there are diagnostic differences in diagnosis depending on sex in the existing literature [23,32], we did not find indication of any distinction between men and women in the first analysis of this study. Similarly, a study carried out in 2008 also found a lack of association in the diagnosis of stress. However, the individual approach of the participants reported higher levels of stress in women. This suggests a potential diagnostic bias of this condition that confirmed this association [23] or differences in how both sexes express and cope with angst [27], which is another decisive aspect of the psychosomatic process.

There were other diagnostic categories in which no significant differences were found either, which may be due to the low number of cases: conversion disorder (n = 35), signs and symptoms related to appearance and behavior (n = 22), PR to occupation and unemployment (n = 18), PR to parenting (n = 7), psychosocial circumstances (n = 25), counseling related to attitude, behavior and sexual orientation (n = 46) or suspicion of abuse, neglect and other types of mistreatment (n = 4). The latter in particular is striking, given the clear correlation that has been shown between women and confirmed abuse, neglect or another mistreatment (aOR = 10.85; 95% CI 2.61–45.14; *p* = 0.001). The group between 25 and 44 years old obtained higher scores, as—according to the analysis—women are up to 14 times more likely to be in this situation (95% CI 1.99–110.37; *p* = 0.009).

Women who have suffered violence largely perceive that their health status is poor or very poor compared to those who have not experience this type of violence. They tend to have a noticeable self-care deficit, especially those who suffered from sexual violence (35.6% of those surveyed) [33]. Therefore, we should be particularly concerned for this sector of the population and associated psychosomatic phenomena, as they are especially susceptible of developing related health issues.

After analyzing the sample data, we also noticed that, although in most of the conditions studied the female sex seems to act as a risk factor for developing health issues, it does not occur in all cases. It was found that being a woman over 65 years of age acted as a protective factor in PR to lifestyle, such us substance use, lack of physical exercise, inadequate diet and eating habits, and high-risk sexual behavior, among others (aOR = 0.26; 95% CI 0.10–0.72; *p* = 0.010). The same was true in the global analysis of non-psychotic mental disorder (aOR = 0.22; 95% CI 0.07–0.70; *p* = 0.001) although no significant differences were seen between the age groups. Additionally, between the ages of 18 and 44, men seem to be at a greater risk to develop signs and symptoms affecting the emotional state (18–24 years old: aOR = 0.34; 95% CI: 0.14–0.80; *p* = 0.013; 25–44 years old: aOR = 0.53; 95% CI: 0.36–0.81; *p* < 0.003) This is consistent with the literature that describes that men and women tend to express and channel their discomforts differently. While they are unconsciously inclined to express anger or take refuge in toxic habits, women tend to repress the emotions generated by conflicts [6]. According to psychosomatic medicine, the latter favors the development of a psychopathological response or physical manifestation without apparent organic origin [34,35]. 

Thus, the aOR indices obtained in this study corroborate that women are significantly more likely to suffer from these conditions. The ages at which the risk increases or decreases are consistent with the literature in terms of the context that may favor one case or the other. It is also worth highlighting that women are considerably more vulnerable to the health problems studied. Consistent with the results obtained, the Board of Andalucía conducted a study from which it was deduced that women are more at risk from developing psychiatric conditions (OR = 1.54) [29]. All this adds to the importance of the analysis by sex and age, since the health of men and women fluctuated due to individual circumstances and the environment.

### Study Limitations

There are several limitations in the current study. The first and clearest is the absence of previous studies that analyze the prevalence of psychosomatic disorders, which precludes a comparison of results and subsequent generalization. Furthermore, this study shows an association between variables, but not cause and effect relationships.

Since this is a retrospective study, the data collection includes binary sex coding, so nonbinary or dissident genders are not considered. Likewise, the variable included in this study is biological sex and, from a biopsychosocial and intersectional approach, analyzing the data from a gender perspective is difficult, given its role as a health determinant.

On the other hand, mental health issues groups notoriously underdiagnosed conditions— especially when the reason for the consultation has marked psychosomatic characteristics—due to lack of training and the inescapable conceptual and clinical ambiguity [4,36]. This implies that, often, these diagnoses are made in institutions outside the National Health System or the ICS. For this reason, we believe we could have obtained more records.

In the case of men, the underdiagnosis may be even more significant, given the impositions imposed by the patriarchal system concerning how to express emotional discomfort. This can cause diagnostic differences between men and women of conditions such as depression or discomfort due to difficulties in expressing emotion learned throughout their life. However, there is no scientific evidence to support that this bias can significantly affect the results [26,30,37]. 

Lastly, information on sociodemographic characteristics of the sample was limited. In order to establish associations supporting the psychosomatic impact of these categories of diagnoses, further studies with a broader scope of the participant’s situation and contextual characteristics are needed.

## 5. Conclusions

The findings of this present study support the existing differential morbidity in psychosomatic disorders according to sex and age.

Notably, more women living in this rural area were diagnosed with these conditions (24.42%) when compared to men (13.1%). This highlights the importance of gender perspective in research, given that obvious differences were observed in the prevalence of illness between men and women.

The female sex showed a higher risk of developing most of the conditions analyzed in this study. These conditions were depression, persistent mood disorder, anxiety, eating disorder, malaise and fatigue, confirmed abuse, neglect and other mistreatment, and problems related to the primary support group. However, female sex and advanced age seem to be protective factors in certain cases. For example, we found that women over 65 years of age were less likely to experience PR to lifestyle, such as substance use, lack of physical exercise, inadequate diet and eating habits, or high-risk sexual behavior, among others. Additionally, between the ages of 18 and 44, men seem to be at a greater risk of developing signs and symptoms affecting the emotional state. These findings may be related to circumstances described by the literature associated with gender issues, such as ineffective emotional management by the male sex, or behaviors learned during socialization linked to a specific gender. It has been shown that age can condition the development of certain issues, generally acting as a risk factor as it increases. 

Thus, the results of this study are relevant for healthcare professionals interested in analyzing the findings intersectionally and designing policies and strategies aimed at improving the detection and treatment of psychosomatic disorders.

## Figures and Tables

**Figure 1 jpm-12-01730-f001:**
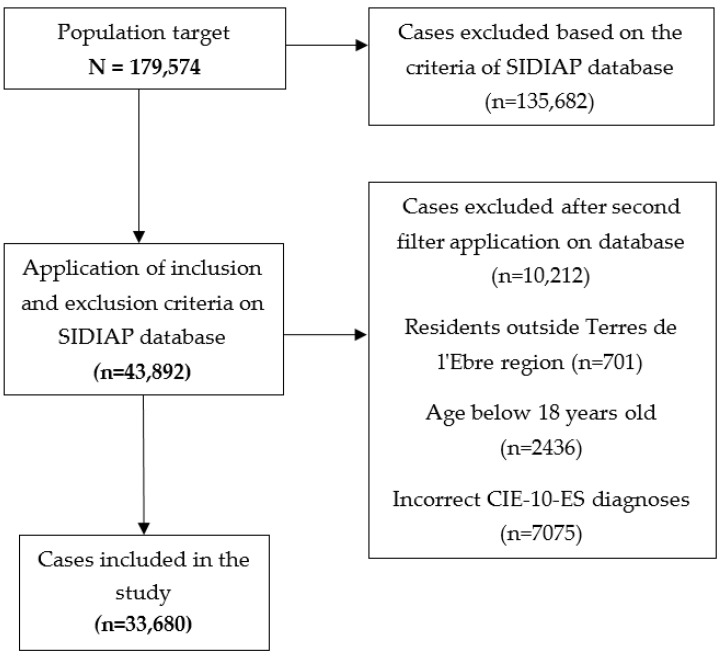
Selection diagram of the cases included in the study.

**Table 1 jpm-12-01730-t001:** Sociodemographic characteristics of the participants.

		Women		Men		
	Total	n	%	n	%	*p*
	33,680	21,772	64.64	11,908	35.36	<0.001
Age						
ALL CASES						
18–24	1390	821	59.06	569	40.94	<0.001 *
25–44	7500	4528	60.37	2972	39.63	<0.001 *
45–64	12,024	7726	64.25	4298	35.75	<0.001 *
More than 65	12,766	8697	68.13	4069	31.87	<0.001 *
MONTSIÀ						
18–24	522	299	57.28	223	42.72	<0.001 *
25–44	2739	1640	59.88	1099	40.12	<0.001 *
45–64	4446	2889	64.98	1557	35.02	<0.001 *
More than 65	4643	3103	66.83	1540	33.17	<0.001 *
BAIX EBRE						
18–24	636	374	58.81	262	41.19	<0.001 *
25–44	3472	2109	60.74	1363	39.26	<0.001 *
45–64	5442	3477	63.89	1965	36.11	<0.001 *
More than 65	5859	4048	69.09	1811	30.91	<0.001 *
RIBERA D’EBRE						
18–24	167	106	63.47	61	36.53	<0.001 *
25–44	879	546	62.12	333	37.88	<0.001 *
45–64	1477	934	63.24	543	36.76	<0.001 *
More than 65	1569	1053	67.11	516	32.89	<0.001 *
TERRA ALTA						
18–24	65	42	64.62	23	35.38	<0.001 *
25–44	410	233	56.83	177	43.17	<0.001 *
45–64	659	426	64.64	233	35.36	<0.001 *
More than 65	695	493	70.94	202	29.06	<0.001 *

Chi-Squared Test. * Significant at *p* < 0.05.

**Table 2 jpm-12-01730-t002:** Association between diagnostic categories and the sex of the study population.

	All Cases	Female	Male	*p*
Depression	8427	5836	2591	<0.001 *
Persistent mood disorder	1213	896	317	<0.001 *
Anxiety	18,311	12,162	6149	<0.001 *
Stress	1945	1296	649	0.059
Conversion disorder	35	28	7	0.057
Somatoform disorder	463	286	177	0.193
Non-psychotic mental disorder	14	4	10	0.005 *
Eating disorder	220	187	33	<0.001 *
Sleep disorder	10,896	6880	4016	<0.001 *
Signs and symptoms affecting the emotional state	751	458	293	0.034 *
Signs and symptoms relating to appearance and behavior	22	11	11	0.151
Malaise and fatigue	1848	1344	204	<0.001 *
Abuse, neglect and other mistreatment, confirmed	37	35	2	<0.001 *
Abuse, neglect and other mistreatment, suspected	4	4	0	0.139
PR to occupation and unemployment	18	11	7	0.754
PR to housing and financial circumstances	736	506	230	0.018 *
PR to the social environment	191	116	75	0.256
PR to parenting	7	5	2	0.707
PR to primary support group, including family circumstances	368	275	93	<0.001 *
Other psychosocial circumstances	25	17	8	0.725
Counseling related to attitude, behavior and sexual orientation	46	36	10	0.053
Contact with healthcare services for other types of medical concern	365	222	143	0.125
PR to lifestyle	59	27	32	0.002 *
PR to difficulty in controlling their life	271	189	82	0.078
PR to caring for a dependent person	836	602	234	<0.001 *

* Significant at *p* < 0.05. PR: Problems related.

**Table 3 jpm-12-01730-t003:** Logistic regression models with grouped diagnostic categories.

		aOR ^a^	95% CI		*p*
Depression					
Sex	Male	1			
	Female	1.28	1.22	1.35	<0.001 *
Age	18–24	1.31	0.93	1.84	0.123
	25–44	1.33	1.17	1.51	<0.001 *
	45–64	1.43	1.30	1.58	<0.001 *
	More than 65	1.61	1.47	1.77	<0.001 *
Persistent mood disorder					
Sex	Male	1			
	Female	1.54	1.36	1.76	<0.001 *
Age	18–24	2.08	1.14	3.80	0.018 *
	25–44	1.55	1.15	2.09	0.004 *
	45–64	1.66	1.33	2.06	<0.001 *
	More than 65	2.09	1.68	2.60	<0.001 *
Anxiety					
Sex	Male	1			
	Female	1.30	1.24	1.36	<0.001 *
Age	18–24	1.63	1.26	2.10	<0.001 *
	25–44	1.38	1.23	1.56	<0.001 *
	45–64	1.44	1.32	1.58	<0.001 *
	More than 65	1.59	1.46	1.75	<0.001 *
Non-psychotic mental disorder					
Sex	Male	1			
	Female	0.22	0.07	0.70	0.001 *
Age	18–24				
	25–44	0.34	0.03	3.78	0.378
	45–64	0.34	0.06	2.07	0.242
	More than 65	0.15	0.02	1.37	0.093
Eating disorder					
Sex	Male	1			
	Female	3.58	2.47	5.20	<0.001 *
Age	18–24	6.11	2.83	13.18	<0.001 *
	25–44	5.31	2.72	10.38	<0.001 *
	45–64	3.51	1.72	7.17	0.001 *
	More than 65	1.63	0.59	4.50	0.343
Sleep disorder					
Sex	Male	1			
	Female	0.82	0.78	0.86	<0.001 *
Age	18–24	0.97	0.66	1.43	0.884
	25–44	0.94	0.82	1.08	0.380
	45–64	1.10	1.01	1.21	0.038 *
	More than 65	1.02	0.93	1.11	0.727
Signs and symptoms affecting the emotional state					
Sex	Male	1			
	Female	0.79	0.68	0.92	0.002 *
Age	18–24	0.34	0.14	0.80	0.013 *
	25–44	0.53	0.36	0.81	0.003 *
	45–64	0.72	0.50	1.03	0.070
	More than 65	0.92	0.76	1.11	0.396
Malaise and fatigue					
Sex	Male	1			
	Female	1.49	1.35	1.66	<0.001 *
Age	18–24	1.14	0.73	1.78	0.565
	25–44	2.09	1.67	2.61	<0.001 *
	45–64	1.53	1.27	1.84	<0.001 *
	More than 65	1.20	1.02	1.43	0.033 *
Abuse, neglect and other confirmed mistreatment					
Sex	Male	1			
	Female	10.85	2.61	45.14	0.001 *
Age	18–24				
	25–44	14.82	1.99	110.37	0.009 *
	45–64	5.54	0.71	43.34	0.103
	More than 65				
PR to housing and financial circumstances					
Sex	Male	1			
	Female	1.06	0.91	1.25	0.443
Age	18–24				
	25–44	0.29	0.13	0.67	0.004 *
	45–64	0.57	0.36	0.91	0.018 *
	More than 65	1.25	1.04	1.49	0.015 *
PR to primary support group, including family circumstances					
Sex	Male	1			
	Female	1.61	1.27	2.04	<0.001 *
Age	18–24	1.69	0.43	6.64	0.450
	25–44	1.52	0.96	2.41	0.077
	45–64	1.83	1.18	2.85	0.008 *
	More than 65	1.48	1.02	2.15	0.039 *
PR to lifestyle					
Sex	Male	1			
	Female	0.47	0.28	0.79	0.004 *
Age	18–24	0.69	0.13	3.69	0.664
	25–44	0.42	0.14	1.25	0.119
	45–64	0.76	0.32	1.76	0.516
	More than 65	0.26	0.10	0.72	0.010 *
PR to caring for a dependent person					
Sex	Male	1			
	Female	1.23	1.05	1.44	0.009 *
Age	18–24				
	25–44	0.98	0.16	5.99	0.979
	45–64	0.68	0.32	1.44	0.315
	More than 65	1.34	1.14	1.58	<0.001 *

^a^ Adjusted Odds Ratio * Significant at *p* < 0.05.

## Data Availability

The datasets used and/or analyzed during the current study are available from the authors on reasonable request.

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
