# Peer review of "Analysis of Psychosomatic Disorders According to Age and Sex in a Rural Area: A Population-Based Study"

_jpm, 2022, doi:10.3390/jpm12101730_

Round 1
Reviewer 1 Report
Thank you for inviting me to review the manuscript. The issues raised by the authors are topical and important for public health. The justification for taking the topic is sufficient. Conclusions are correctly formulated and the discussion is exhaustive. The selection of the test group and the analysis of the results are noteworthy.
Author Response
Dear reviewer,
Thank you very much for your report and comments. We highly appreciate that you find our research interesting and important for the advancement of the health community. It is actually important for us as a researchers on the field but also members of this society, which needs further investigation to improve in mental health knowledge.
Again, very grateful.
Kind regards,
Elisabet Torrubia-Pérez, Silvia Reverté-Villarroya, José Fernández-Sáez and Maria-Antonia Martorell-Poveda.
Reviewer 2 Report
This paper purposed to analyze the prevalence of psychosomatic disorders and their differences according to age and sex in a rural area. I do have some comments as listed below in the order noted.
Comment 1: What is the novelty of this study although several “psychosomatic disorders according to age and sex” program have been proposed earlier?
Comment 2: Please also provide a Table with an association between diagnostic categories and the age of the study population.
Comment 3: Please provide the details why the authors categorized the age group (18–24, 25–44, 45–64, ≥65) in the present study.
Comment 4: In Table 3, please conduct both univariate and multivariate OR (95% C.I.) in the Logistic regression models with grouped diagnostic categories.
Comment 5: Please also add the strengths or significances of the present study in the Discussion section.
Author Response
Dear reviewer,
Thank you very much for your report in order to improve our paper. In the following lines we present our answers to your comments.
Comment 1: What is the novelty of this study although several “psychosomatic disorders according to age and sex” program have been proposed earlier?
Answer comment 1:
Thank you very much for your comment, we are pleased to answer you that previous studies analyse different aspects within this field.
On the one hand, gender studies related to mental health problems are extensive, but these do not usually have the connotation of age as a variable. Introducing age in the analysis allows knowing more accurately the influence of the life cycle (in both genders) in the diagnosis of health problems.
On the other hand, there are not many studies that fit the concept of psychosomatic disorder. Specifically, psychosomatic disorders have an important influence of psychosocial aspects such as age and gender, for this reason in this study we have carried out an analysis contemplating these variables, getting more accurate results.
_____________________
Comment 2: Please also provide a Table with an association between diagnostic categories and the age of the study population.
Answer comment 2:
Thank you very much, we find very interesting your comment. To meet the objective of the study, we considered that the association between diagnostic categories and ages does not provide more relevant information. The reason is that applying OR is already possible to see the association (Table 3) and meeting the proposed aim. Then, trying not to be redundant, we decided to avoid including another table. In order to show you graphically our justification, we attached the table with the association between diagnostic categories and age:
Diagnostic category |
Total |
18-24 |
25-44 |
45-64 |
More than 65 |
p |
Depression |
8427 |
185 |
1451 |
3177 |
3614 |
<0.001* |
Persistent mood disorder |
1213 |
55 |
208 |
422 |
528 |
<0.001* |
Anxiety |
18311 |
897 |
5391 |
7070 |
4953 |
<0.001* |
Stress |
1945 |
136 |
544 |
793 |
472 |
<0.001* |
Conversion disorder |
35 |
8 |
4 |
16 |
7 |
<0.001* |
Somatoform disorder |
463 |
39 |
147 |
169 |
108 |
<0.001* |
Non-psychotic mental disorder |
14 |
1 |
3 |
5 |
5 |
0.954 |
Eating disorder |
220 |
61 |
79 |
59 |
21 |
<0.001* |
Sleep disorder |
10896 |
126 |
1161 |
3456 |
6153 |
<0.001* |
Signs and symptoms affecting the emotional state |
751 |
24 |
97 |
124 |
506 |
<0.001* |
Signs and symptoms relating to appearance and behavior |
22 |
11 |
8 |
0 |
3 |
<0.001* |
Malaise and fatigue |
1848 |
89 |
431 |
606 |
722 |
0.036* |
Abuse, neglect and other mistreatment, confirmed |
37 |
2 |
22 |
11 |
2 |
<0.001* |
Abuse, neglect and other mistreatment, suspected |
4 |
0 |
3 |
1 |
0 |
0.078 |
PR to occupation and unemployment |
18 |
1 |
3 |
9 |
5 |
0.602 |
PR to housing and financial circumstances |
736 |
2 |
26 |
71 |
637 |
<0.001* |
PR to the social environment |
191 |
17 |
17 |
52 |
105 |
<0.001* |
PR to parenting |
7 |
1 |
1 |
3 |
2 |
0.530 |
PR to primary support group, including family circumstances |
368 |
10 |
86 |
112 |
160 |
0.047* |
Other psychosocial circumstances |
25 |
3 |
7 |
11 |
4 |
0.051 |
Counseling related to attitude, behavior and sexual orientation |
46 |
9 |
18 |
11 |
8 |
<0.001* |
Contact with healthcare services for other types of medical concern |
365 |
52 |
102 |
120 |
91 |
<0.001* |
PR to lifestyle |
59 |
6 |
14 |
22 |
17 |
0.085 |
PR to difficulty in controlling their life |
271 |
8 |
32 |
109 |
122 |
<0.001* |
PR to caring for a dependent person |
836 |
2 |
5 |
28 |
801 |
<0.001* |
Chi-Squared Test. * Significant at p<0.05.
_____________________
Comment 3: Please provide the details why the authors categorized the age group (18–24, 25–44, 45–64, ≥65) in the present study.
Answer comment 3:
Thank you for your comment. As we describe at Measures (page 3), in our study, ages were stratified by these age groups according to the classification used by the National Institute of Statistics of Spain (INE) for the analysis of health determinants. INE usually uses this age categorization to present its results. Due to this fact we decided to use the same classification to facilitate the comparison of the results obtained in this work with the data exposed at the national level.
Due to your comment, we have considered interesting to add more information to the paper, which you can consult in the document.
_____________________
Comment 4: In Table 3, please conduct both univariate and multivariate OR (95% C.I.) in the Logistic regression models with grouped diagnostic categories.
Answer comment 4:
Thank you very much, your comment helps us to improve our work. The logistic regression was made with multivariate OR. We adjusted the OR for age. To clarify that we have made minor changes to the document.
_____________________
Comment 5: Please also add the strengths or significances of the present study in the Discussion section.
Answer comment 5:
Thank you very much for your comment. Changes applied to the document.
_____________________
We hope that the modifications respond to your comments correctly. Thank you very much for the work you have done.
Sincerely,
Elisabet Torrubia-Pérez, Silvia Reverté-Villarroya, José Fernández-Sáez and Maria-Antonia Martorell-Poveda.